# Interpretation of Quartz Crystal Microbalance Behavior with Viscous Film Using a Mason Equivalent Circuit

**Sawit Na Songkhla** [1,†] and **Takamichi Nakamoto** [1,2,*,†]

1 School of Engineering, Tokyo Institute of Technology, Yokohama 226-8503, Kanagawa, Japan; sawit@nt.pi.titech.ac.jp
2 Institute of Innovative Research, Tokyo Institute of Technology, Yokohama 226-8503, Kanagawa, Japan
* Correspondence: nakamoto@nt.pi.titech.ac.jp; Tel.: +81-45-924-5017
† These authors contributed equally to this work.

**Abstract:** In odor sensing based on Quartz Crystal Microbalances (QCMs), the sensing film is crucial for both sensor sensitivity and selectivity. The typical response of the QCM due to sorption is a negative frequency shift. However, in some cases, the sorption causes a positive frequency shift, and then, Sauerbrey's equation and Kanazawa's equation cannot be applied to this situation. We model the QCM response with a Mason equivalent circuit. The model approximates a single layer of a uniform viscous coating on the QCM. The simulation of the equation circuit shows the possibility of the positive frequency change when the sorption occurs, which is the situation we find in some of the odor sensing applications. We measured the QCM frequency and resistance using the Vector Network Analyzer (VNWA). The QCMs were coated with glycerol, PEG2000, and PEG20M. To simulate odor exposure, a microdispenser was used to deposit the water. A positive frequency shift was observed in the case of PEG2000, and a negative frequency change was obtained for PEG20M. These results can be explained by the Mason equivalent circuit, with the assumption that when the film is exposed to water, its thickness increases and its viscosity decreases.

**Keywords:** quartz crystal microbalance; vector network analyzer; acoustic impedance; viscoelasticity





## 1. Introduction

We have studied odor sensing systems using an array of Quartz Crystal Microbalances (QCMs) for a long time [1,2]. Although its oscillation frequency typically goes down due to the mass loading effect [3], we sometimes observe a positive frequency shift due to the viscoelastic effect even in the gas phase [4,5] Moreover, we have observed a similar behavior even if we measure the resonance frequency using a network analyzer instead of an oscillator. We would like to understand the behavior of the QCM in the viscoelastic situation.

The working principle of the quartz crystal microbalance sensor is that its wave propagation can be altered by its contacted material. Typically, the frequency shift due to vapor sorption at the sensing film is only mass dependent. However, if the material is soft (small shear modulus value), the mechanical vibration generated from the QCM cannot penetrate deeply into the viscous sensing film, and the upper film surface vibration lags behind the quartz surface vibration. Kanazawa studied a QCM immersed in a liquid medium [6]. Although his analysis agreed with the experimental data, the influence of the viscoelastic film with a finite thickness was not well revealed. Thus, the purpose of the study is to characterize the viscoelastic behavior of the QCM with the film. The viscoelastic property of the sensing film greatly influences the response of the QCM sensor, and its behavior depends strongly on the material properties. The frequency measurement alone is not sufficient when we consider the viscoelastic property.

To obtain more information of the acoustic condition of the QCM surface, the measurement of the electrical impedance around its resonance frequency is necessary. The

modern small sized network analyzers are commercially available, and we embedded it into our measurement system. This method can be interpreted as impedance analysis using a frequency sweep, which is realized by a dedicated circuit or using the network analyzer [7–9]. However, the traditional oscillator is still feasible with liquid loading with a carefully designed circuit [10].

Many approaches to the analysis of the frequency change due to viscous loading have been proposed. One of the proposals is the concept of a mass correcting factor to the frequency change of Sauerbrey's equation for the application of viscous loading [11]. Since viscous loading causes both frequency and resistance change, this viscous loading can be decoupling from the mass loading, since the frequency at half-maximum of viscous loading remains constant [12]. One issue with viscous loading is that the viscosity and density are coupled in the stress equation. A modification to corrugate the QCM surface causes it to trap liquid and act like a pure mass loading, and its comparison with the behavior of the QCM with a smooth surface can reveal the information about the liquid's density [13]. The concept of the Mason equivalent circuit can be extended to a complex shear modulus, making the model prediction more accurate, but the complex parameter is more difficult to tackle [14]. Furthermore, instead of a uniform film, the viscosity of a liquid droplet by simplifying the shape of the liquid yields a relatively accurate result [15]. Some researchers have approached this problem with the model of two elastically bound weights. Their simulation showed that under certain conditions, the frequency shift can be positive [16]. In the recent study on the effect of the shear modulus and viscosity of a thin film, the simulation showed that the viscosity change due to adsorption can significantly affect the sensor response [17]. A more complex coating can also experimentally exhibit both a positive or a negative frequency depending on its molecular conformation [18].

Here, we present an application of a Mason equivalent circuit [19,20], which is based on the acoustic impedance concept. The Mason equivalent circuit derived from the piezo-electric stress equation represents an acoustic impedance. The coupling between the acoustic and electrical impedance makes the conversion between these two variables possible. We omit the complex viscosity and focus on the dissipation due to viscosity alone and reduce the number of parameters, i.e., viscosity (real number), density, and film thickness. Our work focuses on explaining the viscous loading behavior due to the sorption, followed by the confirmation with the experimental results.

## 2. Materials and Methods

### 2.1. QCM Mason Equivalent Circuit

The Mason equivalent circuit consists of 2 acoustic ports, which represent the quartz surfaces on both sides [20]. To simplify the model, one port is shorted, and the other port is treated as a stress-free surface. After several mathematical simplifications, the final equivalent circuitin the from of a Butterworth–Van Dyke network is obtained as shown in Figure 1. We can analyze its equivalent circuit in terms of admittance, and it can be expressed as:

$$Y = \frac{1}{R_1 + j\omega L_1 - \frac{1}{j\omega C_1} + Z_{eL}} + j\omega C_0, \tag{1}$$

where $Z_{eL}$ is the loading from the deposited viscous film. Figure 2 shows a one-dimensional model for the calculation. We consider the case of a viscous thin film, where $\omega$ is the angular frequency, $\mu$ is the shear modulus, and $v_{A1}$ is the shear wave velocity. The particle displacement $u_1$ and the shear stress $T_5$ are given by:

$$u_1 = Ae^{-j(\omega/v_{A1})x_3} + Be^{j(\omega/v_{A1})x_3} \tag{2}$$

and:

$$T_5 = \mu S_5 = \mu \frac{\partial u_1}{\partial x_3} = j\frac{\omega}{v_{A1}}\mu\left(-Ae^{-j(\omega/v_{A1})x_3} + Be^{j(\omega/v_{A1})x_3}\right). \tag{3}$$

*A* and *B* are constant. At the film surface ($x_3 = t + h$), the stress is considered free ($T_5 = 0$). The ratio of *A* to *B* can be solved as:

$$\frac{A}{B} = e^{j(2\omega/v_{A1})(t+h)}. \tag{4}$$

Using Equations (2)–(4), the piezoelectric stress equation can be solved to obtain the acoustic impedance $Z_L$:

$$Z_L = -\frac{T_5}{j\omega u_1}\bigg|_{x_3=t} = \frac{\mu}{v_{A1}}\frac{1 - e^{j(2\omega/v_{A1})}}{1 + e^{j(2\omega/v_{A1})}} = j\rho_L v_{A1} \tan\frac{\omega h}{v_{A1}} \tag{5}$$

where $\rho_L$ is film density and *h* is the film thickness. The shear wave of the viscous film was calculated differently by substituting the shear wave into the velocity in the stress equation. We obtain:

$$\rho_L \frac{\partial^2 u_1}{\partial t^2} = \frac{\partial T_5}{\partial x_3} \tag{6}$$

and:

$$T_5 = j\omega\eta_L S_5, \tag{7}$$

where $\eta_L$ is the film viscosity. Substituting Equation (7) into (6), the wave velocity of the film is:

$$v_{A1} = \sqrt{\frac{j\omega\eta_L}{\rho_L}}. \tag{8}$$

Substituting Equation (8) into (5), we obtain:

$$Z_L = j\sqrt{j\rho_L\eta_L\omega}\tan(\omega h\sqrt{\frac{\rho_L}{j\omega\eta_L}}). \tag{9}$$

Electrical impedance $Z_{eL}$ is expressed as:

$$Z_{eL} = \frac{t^2 Z_L}{4e_{35}^2 S_A}, \tag{10}$$

where *t* is the quartz thickness, $e_{35}$ is the piezoelectric constant, and $S_A$ is the QCM electrode area.

To extract the QCM equivalent circuit component, we used a network analyzer to measure the conductance of the QCM. Then, we applied the curve fitting technique to extract $R_1$, $L_1$, and $C_1$ (in Figure 1) from the conductance curve [21]. This technique was used instead of the conventional motional admittance method, which measures the frequency of maximum conductance [22]. All data points were used to fit to Equation (1) by minimizing the mean squared error; this method is more robust and less susceptible to noise than the conventional motional admittance method.

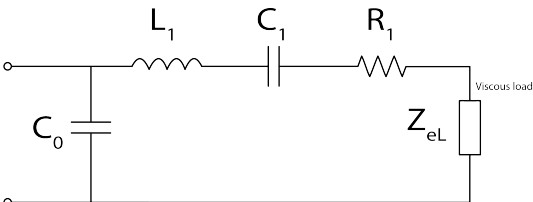

**Figure 1.** The equivalent circuit of the Quartz Crystal Microbalance (QCM), the Butterworth–Van Dyke model with viscous loading.

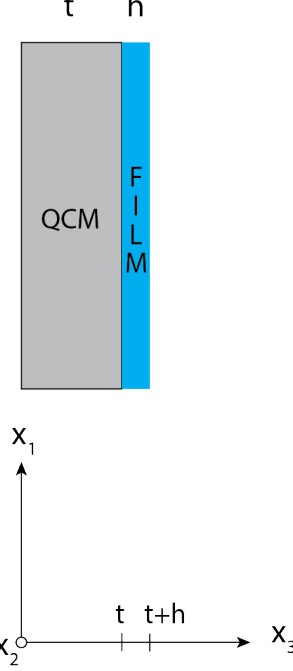

**Figure 2.** Model of the viscous film.

Figure 3 is the simplified one-dimensional model of the film change when exposing it to water. The water sorption causes the overall film thickness to increase and the film viscosity to decrease. This assumption is expected to be able to adequately explain the experimental data so that the Mason circuit model can work.

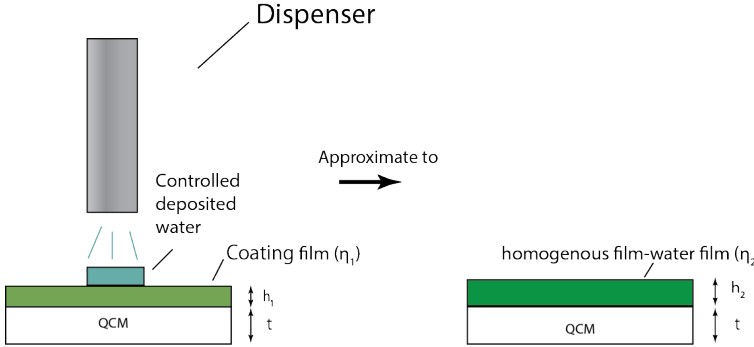

**Figure 3.** Simplified model of the expected film change when exposed to water.

Figure 4a plots the frequency change utilizing Equations (1), (9), and (10), the impedance change due to viscous loading causing the overall impedance change. The QCM parameters used in the simulation are defined as follows: t = 181 um, $e_{35}$ = −0.095, $S_A$ = 1.96 × $10^{-5}$ $m^2$, $L_1$ = 0.0126 H, and $C_1$ = 2.47 × $10^{-14}$ F. These parameters represent the 9 MHz AT-Cut QCM, which was used for every experiment. The frequency change was calculated from the imaginary part of the impedance, and the resistance was calculated from the real part of the impedance. There were 3 variables related to the properties of the coating film: film thickness $h$, film viscosity $\eta_L$, and film density $\rho_L$. Assuming the film density is relatively constant at 1 $g/cm^3$, Figure 4a shows the frequency shift as a function of the deposited film thickness and film viscosity. The same color indicates the same frequency change. Figure 4b shows a similar phenomenon for the resistance value: as the film height increases, the resistance only increases and reaches saturation. This behavior is also true at a high viscosity value if the range of the film height is extended.

For a better visualization, the 3D plots of Figure 4a,b are flattened into the contour plot in Figure 4c,d, respectively. Each line and number on the contour plot represents the

equal value of the frequency and resistance change, similar to the same altitude value when climbing a mountain. Interestingly, we can observe some local minima along the film height change iff we choose a constant viscosity value and progress along the film height (Line A in Figure 4c). First, the frequency decreases, then at a certain range of film thickness, the frequency increases and reaches a constant value. This positive frequency shift due to film thickness change in Figure 4a may explain why, in some cases, the sorption takes place and the positive frequency shift occurs experimentally. Previously, a positive frequency change was thought to be caused by an oscillator behavior. We can analyze this using the dependency of the active impedance of the oscillator circuit upon load change [23]. However, that phenomenon occurred even if we measured the resonance frequency. This can lead us to some explanation about why a positive frequency change occurs simultaneously with a positive resistance change.

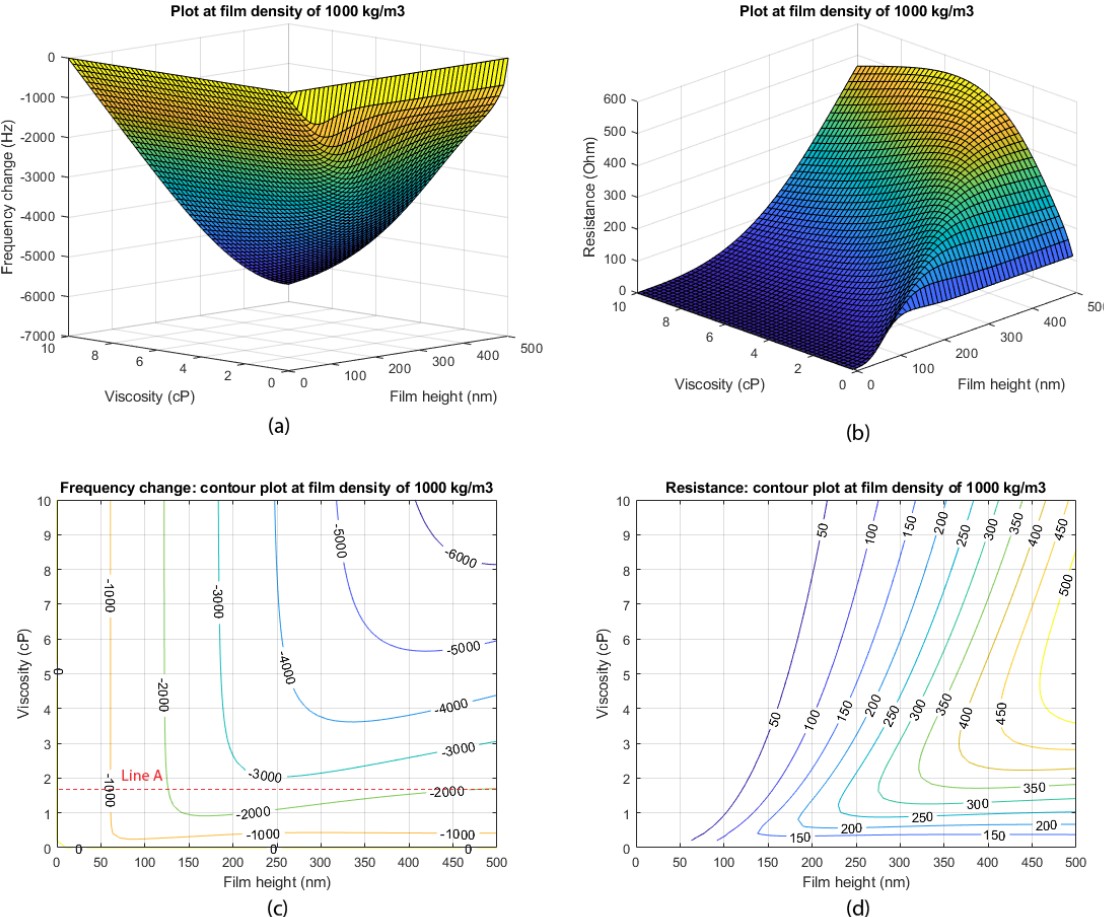

**Figure 4.** 3D plot and contour plot of the frequency shift and resistance at a film density of 1000 kg m$^{-3}$: (**a**) 3D plot of the calculated frequency shift. (**b**) 3D plot of the calculated resistance shift. (**c**) Contour plot of the frequency change. (**d**) Contour plot of the resistance.

### 2.2. Film Coating

The selected film materials were glycerol, PEG2000 (Polyethylene Glycol), and PEG20M. To prepare for coating solution, glycerol was dissolved in acetone (0.01 g/mL), and both PEGs were dissolved in chloroform (0.01 g/mL). Glycerol has a high viscosity, appropriate for testing at the extremely viscous loading. PEG is water soluble, has low toxicity, and is commercially available over a wide range of molecular weights. PEG2000 film is very soft, whereas PEG20M film is rigid. The dip-coating technique was used to coat the QCM (dip coater: VLAST45-06-0100, THK Co., Ltd., Tokyo, Japan). The pull-up speed of the

dip-coater determined the film thickness. The pull-up speed was set to 30 μm/s, whereas the pull-down speed was 300 μm/s. The frequency change after coating of each QCM did not exceed 10 kHz. All experiments were carried out at room temperature. A 9 MHz AT-cut crystal QCM (quartz plate thickness: 181 μm, electrode diameter: 5.05 mm, electrode thickness: 335 nm) with a gold electrode was used throughout the experiments.

### 2.3. Measurement System

The coated QCM was placed inside a 3D-printed holder, as shown in Figure 5a. This was to ensure that the distance and the location of the microdispenser were at the center of the electrode. The microdispenser (INKA2438510H, The Lee Company, CT, USA) [24] operated at 24 V, and the width of its driven pulse was no less than 0.5 ms. The driver circuit is shown in Figure 5c: the ULM2803 is a Darlington transistor driver IC. Figure 5d shows a drawing of the QCM. A single driven pulse of the microdispenser dispensed approximately 4 nL of water. A small sized network analyzer (DG8AQ VNWA, SDR-Kits) was used to measure the S11 parameter from the QCM via a coaxial cable. The measurement software was created with the MATLAB (MathWorks) platform. The frequency was scanned from 8,989,500 Hz to 9,006,500 Hz (a span of 17,000 Hz) with 1000 data points. The measurement time per data point was 2 ms. These measurement parameters were optimized on the basis of the results of our previous experiment [21]. The measurement system diagram is shown in Figure 5b.

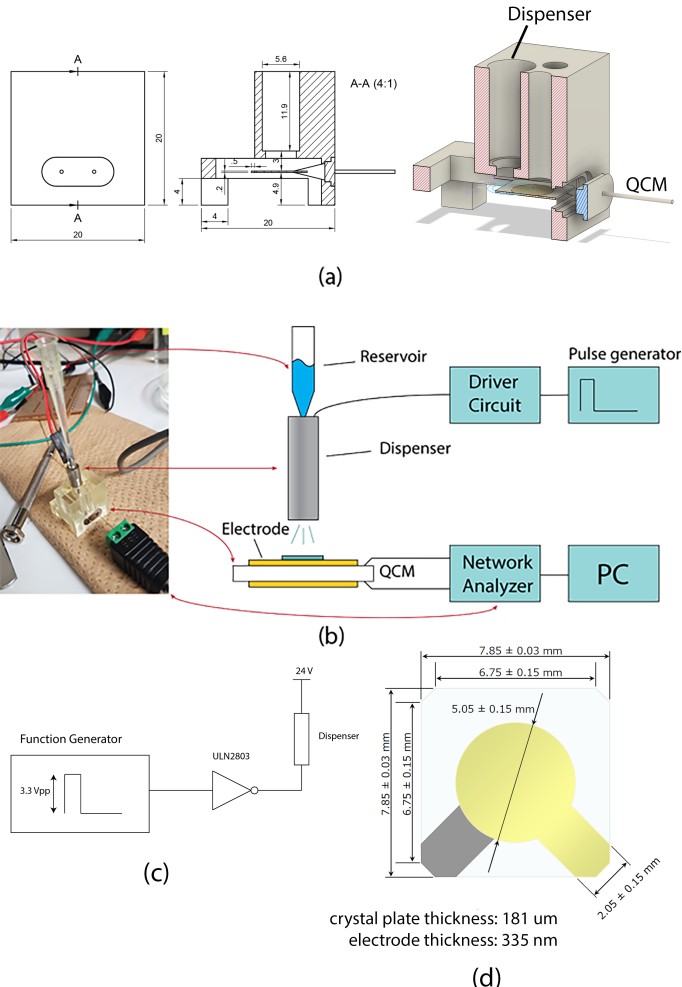

**Figure 5.** (**a**) QCM holder 3D printed design. (**b**) Experimental system setup diagram. (**c**) Dispenser driver circuit diagram. (**d**) Drawing of the QCM.

## 3. Results and Discussion

Experiments were conducted to investigate the response of the QCMs coated with various films. The microdispenser was used to simulate odor exposure; however, we simplified it to use only water deposition. Three different coating films were applied to the QCMs using the dip-coater. Table 1 shows the QCM frequency and resistance change before and after coating. Glycerol had the highest frequency and resistance change, indicating that glycerol is highly viscous.

**Table 1.** Film coating conditions.

| Film Materials | $f_{pre}$ (Hz) | $f_{post}$ (Hz) | $\Delta f$ (Hz) | $R_{pre}$ (Hz) | $R_{post}$ (Hz) | $\Delta R$ (Hz) |
|---|---|---|---|---|---|---|
| Glycerol | 9,004,436 | 8,991,551 | −12,886 | 19 | 740 | 721 |
| PEG20M | 9,001,721 | 8,999,474 | −2246 | 15.92 | 18.28 | 2.36 |
| PEG2000 | 9,001,739 | 8,999,439 | −2300 | 10.98 | 172.22 | 161.2 |

Figure 6 shows the raw QCM data on the water sorption with the glycerol coating. Figure 6a shows the frequency plot, and Figure 6b shows the plot with the adjusted baseline before each exposure. The orange line shows the resistance change. Exposure was in the order of 1 pulse, 3 pulses, 10 pulses, 30 pulses, and 100 pulses. The depositions were repeated three times for each number of pulses. The deposition amount had not saturated the glycerol film yet. Since glycerol is highly viscous, the film height at saturation was very large, as is shown in Figure 4a,c. The dispenser was operated at 1 Hz, yielding 1 s per pulse. There was a positive frequency change during the depositions, but a lingering response after the depositions did not occur. There was a steady-state frequency after each deposition. However, the frequency drift normally occurred independent of water exposure. This was due to the coated film having a large viscosity change even with the small temperature change, and viscous films are more sensitive to this effect. Another cause might be the shape of the film gradually changing due to vapor exposure.

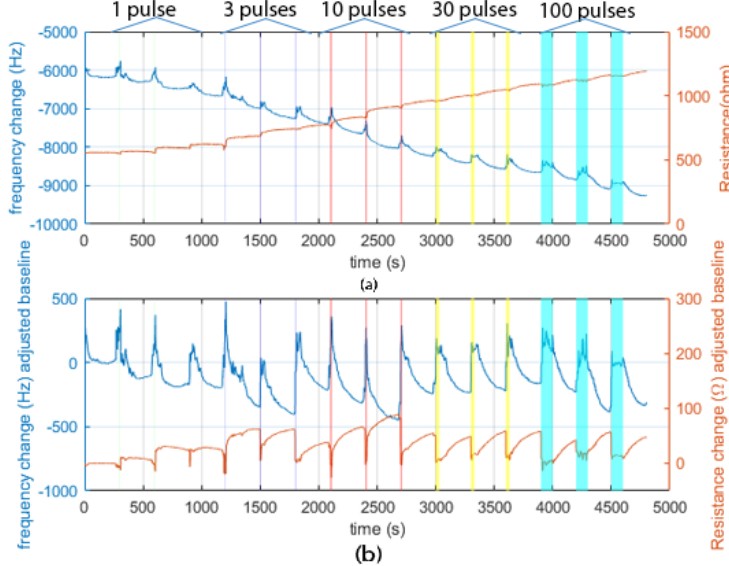

**Figure 6.** Raw QCM data on the deposition of water over glycerol films: (**a**) Frequency change and resistance value. (**b**) Frequency change and resistance value after adjusting the baseline before every deposition.

The experiment was further conducted with Polyethylene Glycol (PEG) film. PEG2000 and PEG20M were chosen as lossy and lossless films, respectively. A lossy material is viscous, and the acoustic energy is consumed at the film; while a lossless material is rigid,

and the acoustic energy is seldom consumed there. Figure 7a,b shows the raw QCM data on the deposition of water over PEG films. The exposure was done in the order of 100 pulses, 300 pulses, and 1000 pulses. The depositions were repeated three times for each number of pulses, similar to Figure 6. Both PEG films had an apparent response after every exposure. PEG20M (Figure 7a) had a negative frequency shift and a positive resistance change. This negative frequency and positive resistance shift response can be interpreted as viscous loading. However, PEG2000 (Figure 7b) had a positive frequency and also a positive resistance. This response cannot be interpreted directly by a combination of Sauerbrey's equation and Kanazawa's equation. Our proposed Mason equivalent circuit had a surface plot with both a positive and negative frequency and resistance change. This makes our model able to describe various types of responses, including this case.

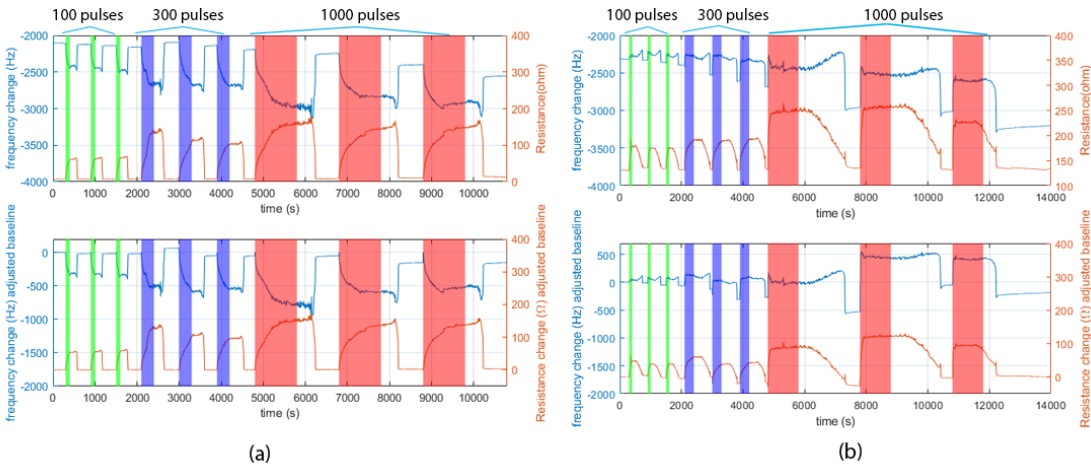

**Figure 7.** Raw QCM data on the deposition of water over PEG film. Both top figures show the raw data; the bottom figures show the data after adjusting the baseline before each deposition: (**a**) Frequency change and resistance value of PEG20M. (**b**) Frequency change and resistance value of PEG2000.

The experimental result of water sorption with glycerol coating can also be roughly explained by the contour plot of the Mason equivalent circuit. The challenge was that the experiment with glycerol had a significant frequency drift independent of the coated film's condition. The blue arrow line in Figure 8 shows the direction of the frequency drift, which in this case tended toward the negative frequency shift. This frequency drift effect did not depend on the coated film and can occur in both positive or negative frequency change. We tried to focus only on explaining the results after the sorption (orange arrows). The orange arrow shows the peak frequency and resistance change after each deposition. To avoid clutter in the plot, we selected only the first deposition out of the three repeated depositions. Our assumption that viscosity decreases as film thickness increases can roughly explain the data. The interesting point is that as the negative frequency change became larger, the resistance became almost constant. This can be explained in the case of 100 pulses. The change might occur along the resistance contour line, but the frequency change still occurs in effect. This may explain why sometimes a frequency change occurs without a resistance change.

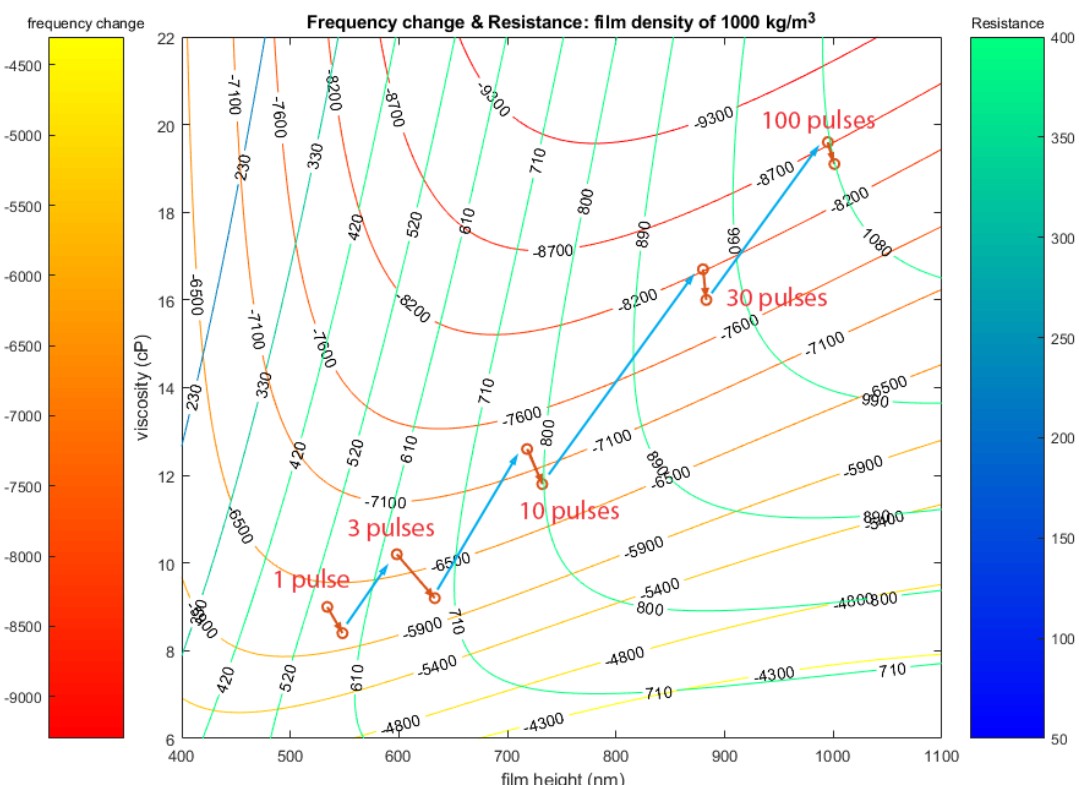

**Figure 8.** Combined contour plot of both resistance and frequency when glycerol film is exposed to water. The orange arrow represents the frequency and resistance change of the coated QCM when exposed to water; the blue arrow indicates the direction of the frequency drift.

Figure 9 shows the combined contour plot of both the frequency shift and resistance. By assuming that the water deposition increases the overall film thickness and decreases the film viscosity by dilution, the direction of change should be from top-left to bottom-right. The frequency shift and resistance data at 100 pulses are drawn for the PEG20M film and the PEG2000 film. PEG20M had a frequency decrease from −2100 to −2400 Hz and had a resistance increase from 7 to 65 Ohm, while PEG2000 had a frequency increase from −2300 to −2200 Hz and had a resistance increase from 130 to 190 ohm. Both films' behavior can be explained by the 3D plot of the Mason circuit. The Mason equivalent circuit can explain the positive frequency change that occurs together with the positive resistance change. This phenomenon cannot be explained using the conventional Kanazawa equation [6].

The data near the bottom right have a tendency toward positive change due to the contour lines of the frequency change being relatively parallel to the film height, making the positive frequency occur when the viscosity decreases. On the contrary, data near the top left have an inverse effect: the frequency shift increases when the film thickness increases. For the resistance change, for almost every case, the resistance increased when there was additional loading. This can be explained by the contour plot. The resistance contour lines always lie diagonally when the viscosity is over 1 cP (viscosity of water = 0.89 cP), which is always the case for almost every coating film. Interestingly, for the thin film (<50 nm), the film behaved as a mass loading only, regardless of the viscosity. We can estimate the film thickness quite accurately by using only the frequency change if the film thickness is thin enough.

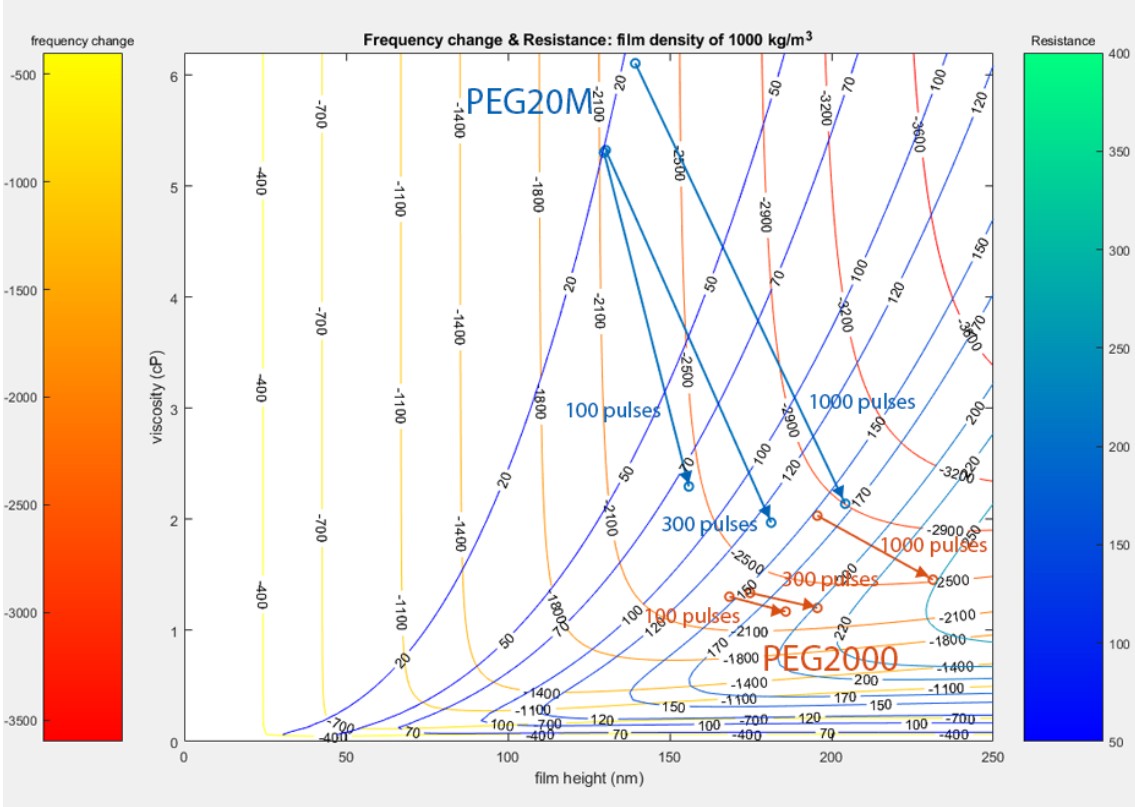

**Figure 9.** Combined contour plot of both resistance and frequency. The arrow line represents the frequency and resistance change of the coated QCM when exposed to 100, 300, and 1000 pulses of water; blue arrow: PEG20M, orange arrow: PEG2000.

## 4. Conclusions

We further explore the Mason equivalent circuit and find its feasibility in predicting a positive frequency change when a certain viscous film is exposed to vapor, to simulate an odorant. The Mason model expresses the impedance of the loading material, which can be converted into a frequency and resistance change for more universal representations.

With a viscous film, the measurement of the frequency alone is not sufficient to detect the material's property changes. The property change due to the viscoelasticity affects both the real and imaginary parts of the impedance; to measure both parts, the measurement of both the frequency change and resistance change is necessary. The Mason equivalent circuit provides an analytical equation and is valid for a wide range of viscosities.

Even with a simple assumption of film thickness increase and viscosity decrease when sorption takes place on the viscous film, our equation can roughly explain the scenarios of both a lossy film (PEG2000) and a low loss film (PEG20M). For the glycerol film, if we consider the peak signal change after the depositions, the result can also be explained. Since the derived model is basically equivalent to a transmission line model, it can also be extended to a multi-layer model or even treat the water deposition as an additional layer. Although this study focuses only on the explanation of the positive frequency shift, the sensitivity of a QCM coated with a viscous film will be investigated in the future. The raw data of QCM on the deposition of water over glycerol, PEG20M and PEG2000 coatings are available in Tables S1–S3, respectively.

**Supplementary Materials:** The following are available online at https://www.mdpi.com/2227-904 0/9/1/9/s1. Table S1: Raw QCM data on the deposition of water over glycerol film, Table S2: Raw QCM data on the deposition of water over PEG20M film, Table S3: Raw QCM data on the deposition of water over PEG2000 film.

**Author Contributions:** Conceptualization, T.N.; methodology, T.N. and S.N.S.; software, S.N.S.; validation, S.N.S.; formal analysis, S.N.S.; investigation, S.N.S.; resources, T.N.; data curation, S.N.S.; writing, original draft preparation, S.N.S.; writing, review and editing, T.N.; visualization, S.N.S.; supervision, T.N.; project administration, T.N.; funding acquisition, N/A. All authors read and agreed to the published version of the manuscript.

**Funding:** This research received no external funding.

**Data Availability Statement:** The data presented in this study are available in the supplementary material.

**Acknowledgments:** This paper and the research behind it would not have been possible without the support of Japan International Cooperation Agency (JICA), which has provided generous support through the "Innovative Asia" scholarship program.

**Conflicts of Interest:** The authors declare no conflict of interest.

## Abbreviations

The following abbreviations are used in this manuscript:

QCM    Quartz Crystal Microbalance
PEG    Polyethylene Glycol

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
