# Peer review of "Interpretation of Quartz Crystal Microbalance Behavior with Viscous Film Using a Mason Equivalent Circuit"

_chemosensors, doi:10.3390/chemosensors9010009_

Round 1
Reviewer 1 Report
This manuscript proposed a simulation and modelling method for QCM response using Mason equivalent circuit which could account for positive frequency shift in some odor sensing applications. The work is well-organized, and the paper can be published after some revisions. The comments are as follows:
- The keywords may not be very appropriate. The manuscript focuses on simulated experiments, corresponding modeling and analysis, it does not directly and mainly discuss gas sensing. So, some appropriate keywords need to be extracted.
- Please discuss more recent publications in this field, and redundant publications such as references [5-8] should be condensed.
- Please provide more information of QCM such as the provider, picture, size and electrode thickness of QCM.
- The conclusion of Figure 4b in line 102 seems to be not rigorous enough. In parts of the viscoelastic range (such as 8-10cP), the trend is still an accelerated upward trend.
- Why didn’t Figure 8 provide the results of all pulses like Figure 9? Besides, to avoid clutter in plotting we select only first deposition out of the three repeated 187 deposition. But the repeatability cannot be verified. It is recommended to display all the results using error bars or other methods.
- The abstract indicates that the positive frequency shift is the focus of the manuscript. But only PEG2000 has a positive frequency shift. Is this phenomenon related to some specific materials? And what
- What is lossy film and lossless film, and the difference between their characteristics should be explained briefly.
- The positive frequency shift in Figure 8 is the most important point of the manuscript, so could you provide more data like Figure 9 and draw them in the Figure 8?
- The contents of Figure 6 and Figure 9 seem to be the same. So could you explain why Figure 9 is placed at the end instead of after Figure 6.
- In line 150, Could you provide more explanations or speculations for the cause of frequency drift as well as the increase of viscosity at the same time, such as temperature and system stability.
- There are some writing and grammatical errors in the manuscript such as line 17 “mass mass” and line 30 “the its”. It is recommended to check the manuscript carefully.

Author Response
- The keywords may not be very appropriate. The manuscript focuses on simulated experiments, corresponding modeling and analysis, it does not directly and mainly discuss gas sensing. So, some appropriate keywords need to be extracted.
The keywords were revised to quartz crystal microbalance, vector network analyzer, acoustic impedance and viscoelasticity.
- Please discuss more recent publications in this field, and redundant publications such as references [5-8] should be condensed.
We condensed our references and reviewed additional recent publications at line 51.
- Please provide more information of QCM such as the provider, picture, size and electrode thickness of QCM.
The drawing of QCM was added to figure 5(d) and the information of QCM were added to section 2.2 line
130.
- The conclusion of Figure 4b in line 102 seems to be not rigorous enough. In parts of the viscoelastic range (such as8-10cP), the trend is still an accelerated upward trend.
The additional explanation was added to line 105.
- Why didn’t Figure 8 provide the results of all pulses like Figure 9? Besides, to avoid clutter in plotting we select only first deposition out of the three repeated 187 deposition. But the repeatability cannot be verified. It is recommended to display all the results using error bars or other methods.
We added extra arrow for data of 300 and 1000 pulses similar to figure 9 (previous figure 8). Please see figure 7 for the raw data plot for the repeatability of frequency after exposure to water.
- The abstract indicates that the positive frequency shift is the focus of the manuscript. But only PEG2000 has a positive frequency shift. Is this phenomenon related to some specific materials? And what
For the positive frequency shift to occur, the initial viscosity and film height have to be within the lower right region of the contour plot (current figure 9), within this region the frequency change contour line is relatively parallel to the film thickness axis (x axis). The water deposition over film causes the film thickness to increase but viscosity to decrease, this moves the frequency to positive direction.
- What is lossy film and lossless film, and the difference between their characteristics should be explained briefly.
The explanation of lossy and lossless film were added to section 3 line 163.
- The positive frequency shift in Figure 8 is the most important point of the manuscript, so could you provide more data like Figure 9 and draw them in the Figure 8?
We revised the figure as answered in question 5.
- The contents of Figure 6 and Figure 9 seem to be the same. So could you explain why Figure 9 is placed at the end instead of after Figure 6.
We had swapped the order of figure 8 and figure 9 to improve the consistency as suggested.
- In line 150, Could you provide more explanations or speculations for the cause of frequency drift as well as the increase of viscosity at the same time, such as temperature and system stability.
The cause is the coated film can have a large viscosity change in spite of the small temperature change since viscous films are more sensitive to this effect. Another cause might be the shape of the film might gradually change due to exposure to water. The explanation was added to line 158 .
- There are some writing and grammatical errors in the manuscript such as line 17 “mass mass” and line 30
“the its”. It is recommended to check the manuscript carefully.
Our apologies for the typo. We have corrected these errors.

Reviewer 2 Report
In this work the authors report a computational model based on a Mason equivalent circuit to predict the change in frequency and change in resistance response at a QCM presenting a thin viscous film. They collected experimental data for the hydration of three different films that were deposited on the QCM surface (glycerol, PEG2000, and PEG20m) and used a network analyzer to extract the relevant circuit parameters. They compared the experimental results to the predictions made by the computational model to show the effectiveness of the computational model. The experiments were appropriately carried out and the comparison to the computational model is interesting. I believe these results should be of interest to the readers of Chemosensors and to the QCM community. A few items should be addressed prior to publication:
- There are some proof-reading and grammatical issues throughout the text that should be improved.
- Figure 6 – Is there any concern of water saturation after 3x 100 pulses or 3x 1000 pulses? Is there any concern that the response decreases after repeated pulses of water? The amount of water delivered is very small, so this may not be a concern.
- Figure 7b – These results are interesting as the authors state. The increase in frequency and decrease in resistance may be due to changes in the film viscosity as discussed in the manuscript. It is apparent that the rise in frequency occurs after the pulse period during the equilibration time. Is there any concern that the rise in frequency is due to a physical phenomenon such as evaporation of the water off of or out of the thin layer? The response appears to decrease with each replicate of pulses as well. Could this be due to sensor saturation as mentioned above?
- Pg 8, line 169 – The authors state that the film behavior can be explained by the 3-D plot of mason circuit. I’m not certain that the computational model explains the results, but rather models or allows for prediction of the result. If the model is explaining the phenomenon, then could the authors be clear about what the explanation is?
- Pg 9, line 187 – “Our assumption that viscosity decreases as film thickness increases decrease can roughly explain the data.” Please clarify “increases decreases”.
- Figure 9 – Is the QCM drift important in this case once the data is baseline corrected as shown in Figure 7?
Author Response
- There are some proof-reading and grammatical issues throughout the text that should be improved.
- We have thoroughly checked the manuscript and revised it.
2. Figure 6 – Is there any concern of water saturation after 3x 100 pulses or 3x 1000 pulses? Is there any concern that the response decreases after repeated pulses of water? The amount of water delivered is very small, so this may not be a concern.
- We have not deposited water with its amount to saturate glycerol film yet. Since glycerol is highly viscous, the film height at the saturation is very large as is shown in figs. 4(a) and (c). This explanation was also added to line 154.
3. Figure 7b – These results are interesting as the authors state. The increase in frequency and decrease in resistance may be due to changes in the film viscosity as discussed in the manuscript. It is apparent that the rise in frequency occurs after the pulse period during the equilibration time. Is there any concern that the rise in frequency is due to a physical phenomenon such as evaporation of the water off of or out of the thin layer? The response appears to decrease with each replicate of pulses as well. Could this be due to sensor saturation as mentioned above?
- The frequency rise after the deposition period may be due to the shape of the film changes as it adsorbs water because the film material may spread out further from the center and the overall mass loading may decrease. The response decrease for each replicate deposition may be caused by gradual increase in amount of the water accumulated inside the film even if the water somewhat evaporates.
4. Pg 8, line 169 – The authors state that the film behavior can be explained by the 3-D plot of mason circuit. I’m not certain that the computational model explains the results, but rather models or allows for prediction of the result. If the model is explaining the phenomenon, then could the authors be clear about what the explanation is?
- The mason equivalent circuit can explain the positive frequency change that occurs together with positive resistance change. This phenomenon cannot be explained using conventional Kanazawa’s equation. This explanation was also added to line 194.
5. Pg 9, line 187 – “Our assumption that viscosity decreases as film thickness increases decrease can roughly explain the data.” Please clarify “increases decreases”.
- Our apologies for the typo. The sentence is “Our assumption that viscosity decreases and film thickness increases can roughly explain the data”.
6. Figure 9 – Is the QCM drift important in this case once the data is baseline corrected as shown in Figure 7?
- The baseline drift does not affect the use of this model, since the absolute frequency change (calculated from bare QCM) is used on the plot.

Reviewer 3 Report
The manuscript focused on positive drift of QCM to present a model of the QCM response with Mason equivalent circuit. The model approximates a single layer of a uniform viscous coating on to the QCM. The study is interesting for readersto understand why and how does? But the authors neglect a fact that interaction between gas and coating. I suggest strongly the authors add some relative literatures in P1 line 19 after” we sometimes observed positive frequency shift due to viscoelastic effect even in the gas phase, such as NH3 Sensing Mechanism Investigation of CuBr: Different Complex Interactions of the Cu+ Ion with NH3 and O2 Molecules,Zhang Yuan; Xu Pengcheng; Xu, Jiaqiang; Li Hui; Ma Wenjie,J. Phys. Chem. C,2011,115(5): 2014-2019, and so on. In addition, the cited references are too sufficient to show your research value, I suggest you add the citions for readers after P1 line 18 after Although its oscillation frequency typically goes down due to mass mass loading effect[3]

Author Response
Reviewer 3
The manuscript focused on positive drift of QCM to present a model of the QCM response with Mason equivalent circuit. The model approximates a single layer of a uniform viscous coating on to the QCM. The study is interesting for readers to understand why and how does? But the authors neglect a fact that interaction between gas and coating. I suggest strongly the authors add some relative literatures in P1 line 19 after” we sometimes observed positive frequency shift due to viscoelastic effect even in the gas phase, such as NH3 Sensing Mechanism Investigation of CuBr: Different Complex Interactions of the Cu+ Ion with NH3 and O2 Molecules,Zhang Yuan; Xu Pengcheng; Xu, Jiaqiang; Li Hui; Ma Wenjie,J. Phys. Chem. C,2011,115(5): 2014-2019, and so on. In addition, the cited references are too sufficient to show your research value, I suggest you add the citions for readers after P1 line 18 after Although its oscillation frequency typically goes down due to mass mass loading effect[3]
- Thank you for the advice we have fixed the typo and have included additional references per your advice.
Round 2
Reviewer 1 Report
After revised, I agree the publication of this paper.
Reviewer 2 Report
The authors have addressed all reviewer questions. This manuscript is acceptable for publication.
Reviewer 3 Report
The revisions are satisfactory, I recommend its publication.